# Development and Implementation of an Integrated Framework for Undergraduate Pharmacy Training in Maternal and Child Health at a South African University

**DOI:** 10.3390/pharmacy9040163

**Published:** 2021-10-08

**Authors:** Elizabeth Oyebola Egieyeh, Angeni Bheekie, Mea van Huyssteen, Renier Coetzee

**Affiliations:** School of Pharmacy, University of the Western Cape, Private Bag X17, Bellville, Cape Town 7535, South Africa; abheekie@uwc.ac.za (A.B.); mvanhuyssteen@uwc.ac.za (M.v.H.); recoetzee@uwc.ac.za (R.C.)

**Keywords:** curriculum, integrated maternal and child health care framework, undergraduate pharmacy students, knowledge, skills and attitude, pharmacy education, South Africa

## Abstract

The South African Pharmacy Council (SAPC) regulates undergraduate pharmacy education and pharmacy practice. The SAPC Good Pharmacy Practice manual describes the role of pharmacists in maternal and child health (MCH) in line with the recommendation of international health regulatory bodies. However, baseline study findings in 2017 supported literature from around the world that indicated a need for curriculum review and integration to address the knowledge and skills gap in pharmacists’ MCH training. This paper describes the development and implementation of an integrated framework for MCH training across the four years of a Bachelor of Pharmacy program. The intervention included didactic lectures, skills practical on infant growth assessment, and an experiential learning component at primary health care clinics and pharmacies. Knowledge and skills assessment on contraception, maternal and antenatal care, and neonatal and child care were carried out pre, eight weeks post, and two years post intervention using the same questionnaire. ANOVA and post hoc analyses showed that participants’ knowledge and skills increased post intervention but decreased significantly two years later except in contraception where students experienced longitudinal integration of the MCH component. Generally, participants performed above the university average except in maternal and antenatal care.

## 1. Introduction

The delivery of maternal and child health (MCH) services in South Africa is undertaken at the primary health care (PHC) level [1]. As one of the quadruple burdens of disease in the country, the high maternal (121 per 100,000 in 2015) and under-five mortality (42 per 1000 live births in 2016) ratios require that all health care professionals who participate in MCH care delivery are effectively trained to deliver the required health care services to this group of patients [2,3]. The high rate of unplanned and teenage pregnancies further contributes to maternal and child mortality upsurges from unsafe abortions and the increased risk of complications for both mothers and babies [4]. The high mortality ratio is attributed to a lack of access to skilled health care professionals and quality health care services [5].

Pharmacists’ roles as providers of indirect obstetric care in MCH are documented in the International Pharmaceutical Federation’s (FIP) statement of policy on the effective utilization of pharmacists in improving maternal, neonatal, and child health [3,6]. The delivery of care cuts across the continuum of MCH care and is in line with World Health Organization’s (WHO) suggested interventions to low- and middle-income countries with high burdens of mortality [7]. 

The regulation of undergraduate pharmacy education (Bachelor of Pharmacy) and practice in South Africa is overseen by the South African Pharmacy Council (SAPC) [8]. The SAPC has competency standards to guide undergraduate education and training in line with the FIP global competency framework (2012), which are relevant to MCH at the PHC level (Table 1) [9]. The minimum professional standards required to provide MCH services at the PHC level usually offered in community pharmacies are prescribed in the SAPC’s Good Pharmacy Practice manual [8]. The manual emphasizes the need for adequate training, knowledge, and skills. The importance of understanding the physical facility, equipment, procedures, documentation, record keeping, and confidentiality necessary to render these services is also highlighted. The MCH services that may be rendered by pharmacists include immunization, pregnancy tests, nutrition in pregnancy, baby and child health care, and reproductive health services. Pharmacist can offer comprehensive reproductive health services if the necessary postgraduate training has been obtained and registered with the SAPC. However, pharmacists are allowed to initiate emergency postcoital contraception without additional training. As such, undergraduate training is expected to provide entry level pharmacists with the competencies required to render MCH services expected at the PHC level.

Global reforms in health professional education are directed towards strengthening the health care system through collaboration with health education institutions [10]. In resource constrained developing countries such as South Africa, the reform needs to be hinged on interventions that meet the health care needs of the majority of the population [11]. Needs based pharmacy education as proposed by the FIP Educational Initiative (FIPEd) aims to ensure that pharmacy schools are socially accountable; that practice and science are evidence-based; and pharmacy graduates have the competencies to provide the required health care services for their communities [12].

Unfortunately, traditional pharmacy department or faculty generally works along disciplines that inform curriculum development [13]. This leads to compartmentalized and fragmented student learning where the link between the topics taught by each discipline is left to the student to decipher. This is in contrast to an integrated curriculum where the teaching method is contextual, applied, nestled, connected, and dispersed longitudinally to motivate students and enhance learning, knowledge retention, and application [14,15,16,17]. Integration also helps students to think critically and become problem solvers [18,19,20]. Curriculum integration in MCH has been shown to be more effective when linked with community-based learning and MCH primary health care facility engagements, which exposes students to community issues and helps to them to develop preventive thinking abilities [14,21,22].

This study follows on from an evaluation conducted in 2017 of final year pharmacy students’ knowledge and skills in reproductive, maternal, new-born, and child health care at a South African university following a traditional, fragmented curriculum content exposure [3]. As a result of the evaluation, curriculum revision and longitudinal integration of MCH module content was recommended. This paper explains the development and implementation of a framework for integrated MCH curriculum content. It explores the effect of the longitudinal integration on students’ knowledge and skills acquisition and retention to inform teaching and learning improvement. The study is the only one at present to our knowledge that has developed and implemented a framework for MCH training in undergraduate pharmacy education in South Africa.

## 2. Materials and Methods

### 2.1. Study Design

This was a longitudinal cohort study of undergraduate pharmacy students who were exposed to an integrated maternal and child health program in all four years of the curriculum from 2016 to 2019, with the main content concentration in the second year. The knowledge and skills of the participants in their second (2017) and fourth (2019) years of study was assessed.

### 2.2. The Setting

The School of Pharmacy, University of the Western Cape is the only pharmacy school in the Western Cape province of South Africa. The school has four main academic disciplines: pharmaceutical chemistry, pharmacy practice, pharmaceutics, and pharmacology and clinical pharmacy, which integrates the experiential learning component, one of which is referred to as service learning in pharmacy (SLiP). A modular system of curriculum content delivery is adopted by each discipline. In 2017, an intervention was introduced at the second-year level to improve the MCH training through curriculum content enhancement and integration in response to government initiatives to reduce the mortality rate in the country. Existing curriculum content with a bearing to the topic was identified and new content was introduced to develop a framework (Table 1) that would enhance student learning and improve their competence using multiple teaching methods.

### 2.3. Developing the Framework

The MCH exposure began in the second semester of the first year of study in the environmental and nutritional pathology lectures of the Introduction to Pharmacology and Clinical Pharmacy (PHC 123) module. One of the focus areas of the lecture was diarrheal disease, a communicable disease associated with neonatal and child mortality in South Africa [22]. An environmental health SLiP programme was attached to the module where students visited underserved communities accompanied by City of Cape Town environmental health practitioners to have real-life exposure to the social determinants of health and the risk factors for disease, as taught in the lecture. In addition, students were guided by qualified and authorized faculty staff to prepare dry mixtures of oral rehydration salt and sugar prepacks used in the treatment of diarrheal disease [23].

In the second year of study, an infant growth assessment practical (Figure 1) was introduced into the clinical skills training in the Pharmacology and Clinical Pharmacy module (PHC 213) to prepare students for the MCH program in the second semester (PHC 223). The MCH program comprised of an average of 15 h of didactic lectures that concluded the on-campus training. I ran concurrently with the SLiP-MCH component, which took place at the MCH units of public PHC facilities in the Cape Town Metropole. The lectures covered the continuum of care in MCH, such as contraception (part 1 included a contraceptive products demonstration practical), preconception, pregnancy, antenatal, post-natal, and infant care, all framed within pharmacists’ scope of practice. The lecture content in PHC 123 and PHC 223 was taught by the same faculty staff member (EOE). The SLiP-MCH clinic sessions took place from the second week of the semester to the eighth week. The class was split into two groups for the experiential learning sessions. Each group spent a total of 9 h for three weeks (3 h for 3 consecutive Fridays from 8:00–11:00 a.m.) at the facilities. They engaged in health educational activities to encourage and educate pregnant teenage girls on contraception, health, and well-being promotion activities for mothers and infants, and counselled mothers on medicine administration for nevirapine, vitamin A drops, oral rehydration solution (ORS), and deworming agents. In addition, they took part in infant growth assessments, charted and interpreted information in the infant ‘road to health’ booklet under the direct guidance of facility nurses. As part of the school’s social accountability drive, the salt and sugar dry powder oral rehydration prepacks prepared by the first-year students in PHC 123 were delivered to the facilities by the second-year students during their MCH service sessions [5,16]. In the middle of the following week, a reflection session was held where faculty members helped students tease out the work-based learning that occurred at the facilities [24]. Students were required to submit an individual reflection report which rounded off the second-year MCH programme.

In the third year of study (2018), an MCH component was introduced into the July holiday externship program coordinated by the pharmacy practice discipline (PPR 324). Students were required to carry out self-initiated patient activities similar to the second year SLiP-MCH activities at retail pharmacies or health care facilities of their choice during the mid-year holiday externship as part of the 400 h of work-based learning required by the SAPC [25]. 

The MCH program was finally rounded off in the first semester of the fourth year (2019) with the concluding part of the contraception lectures [3].

### 2.4. Questionnaire Development

The questionnaire (see Appendix A for the questionnaires) contained 34 items that totaled 46 marks. The items consisted of 18 multiple-choice (one mark per question) and 16 short answer questions (one mark per response) which counted for 28 marks. The questionnaire was divided into three main sections A, B, and C. Section A comprised of participants’ demographic details, which included age, gender, participants’ first language, previous exposure to any MCH content, locum (a person who substitutes for another person from the same profession to temporarily fulfil their duties) experience, and parental status. Section B (28 items) was the knowledge section, it had three sub-sections: reproductive and sexual health, centered on contraception (9 items, 9 marks); maternal and antenatal care, which assessed participants preconception and pregnancy care knowledge (10 items, 17 marks); and neonatal and child care, which covered infant care and nutrition, childhood diseases and immunization (9 items and 12 marks). Section C (6 items, 8 marks) assessed participants infant growth assessment skills and knowledge in a written format. Participants required a score of 50% in each knowledge subsection and the skills section of the questionnaire to pass the assessment. This was in alignment with the university’s 50% pass mark.

### 2.5. Study Participants

In 2017, only second year pharmacy students who had taken part in the infant growth assessment practical in PHC 213 and were registered for the PHC 223 module could participate in the study. A convenience sample of 97 students consented to participate in the study. 

The inclusion criteria for participating in the fourth-year study in 2019 required students to be registered at the university, to have completed the 2018 MCH component of the externship and to have participated in the second-year MCH intervention and study assessments. A purposive sample of 47 students agreed to continue with the study. This paper focuses on the 47 students who completed the three assessments. 

### 2.6. Recruitment and Data Collection

Participants recruitment and data collection was done in two phases.

Phase 1: This phase involved 2017 second year pharmacy students who were registered for PHC 223 module (Table 1). Participant recruitment was carried out in the first week of the second semester during a normal class time and in a lecture venue allocated to the module (PHC 223). This was undertaken before the MCH didactic lectures were delivered. The researcher (EOE) explained the purpose of the study to the class and provided interested students with the study information sheet and consent form to complete and return to the researcher. The pre-intervention assessment questionnaire was handed out to a convenient sample of 97 consenting students (Figure 2) to complete in other to assess their baseline knowledge. 

On completion of the didactic lectures, contraceptive products demonstration practical, SLiP-MCH clinic, and reflection sessions eight weeks after the pre-intervention assessment, a post intervention assessment was carried out using the same questionnaire and procedure. Participants were informed that they would be requested to participate in a third assessment in their fourth year of study in 2019.

Phase 2: The third assessment was carried out in 2019 at the beginning of the second semester of the participants fourth year of study using the same questionnaire (Figure 2). An email was sent to the fourth-year class via the university’s online platform to inform them of the assessment. At a pre-arranged lecture venue and time slot, students were invited to complete the consent form and participate in the assessment. A purposive sample of 47 students who had participated in the 2017 assessment consented to participate in the assessment and complete the questionnaire.

### 2.7. Data Evaluation and Analysis

Three research students graded the completed questionnaires using a structured memorandum. Two other research students moderated all the graded questions. The marks were captured on an Excel spreadsheet by a research assistant and exported into the Statistical Package for Social Sciences (SPSS) version 26 for analysis [26]. One sample t-test was used to compare participants’ mean scores with the average score of 50% to determine the pass rate. One-way repeated-measures analysis of variance (ANOVA) and post hoc tests were used to determine the difference in participants mean scores in the three assessments. Stepwise linear regression analysis was used to predict the effect of participants demographics on knowledge and skills acquisition and retention. Cronbach’s alpha was calculated to quantify the test reliability.

## 3. Results

### 3.1. Participants Demographics

Forty-seven (47) participants completed all three assessments in the second and fourth years of their study. Most of the participants were between 18–30 years old (93.6%), and 36 (77%) were female (Table 2). None of the participants were parents in the 2017 pre and post intervention assessments. The first language of communication for 24 (51%) participants was English, while the others (49%) had an African language which included Afrikaans. Ten (21%) of the second-year participants had some knowledge of MCH prior to their exposure to the school’s curriculum content through work-based experiences, information from parents or workshops or personal experiences. In the fourth year of study, one participant had done locums for more than two years, 11 others for one to two years, while 35 (75%) had never did locum.

### 3.2. Participants’ Knowledge and Skills Assessment

Results of participants performance in the three assessments was compared to the university’s stipulated pass mark of 50% in Table 3. In the 2017 pre-intervention assessment, only the knowledge components were assessed. Participants’ mean scores were significantly above the pass mark of 50%, except in neonatal and child care, where the mean score was significantly below 50% (43%, *p* = 0.016). The overall average score of the participants was above 50%.

In the 2017 and 2019 post intervention assessments, participants’ mean scores were significantly above the 50% pass mark except in the 2019 infant growth assessment skills section where the mean score was insignificantly above the pass mark (*p* = 0.187). The mean score difference between the MCH components and the pass mark of 50% was higher the 2017 assessment.

The difference in participants’ mean scores in the three assessments are compared in Table 4. The one-way repeated-measures ANOVA showed that there was a significant difference in participants mean knowledge and skills scores across the three assessments (Wilks’ Lambda = 0.31, F (2, 45) = 51.0, *p* = 0.0005, multivariate partial eta squared = 0.694). A post hoc pairwise comparison using the Bonferroni correction was used in the analysis.

Statistically significant (*p* < 0.05) increases in participants mean scores was observed between the 2017 pre- and post intervention assessments for all knowledge components evaluated in the study. The biggest increase was observed in neonatal and child care (33.2%) which had the least baseline mean score, while the smallest increase was observed in maternal and antenatal care (13.4%) which had the highest baseline mean score.

The mean difference between the 2017 and 2019 post intervention assessments showed significant decreases (*p* < 0.05) in knowledge and skills retention in all the MCH components except in reproductive and sexual health where the decrease was not significant (*p* = 0.283). The biggest decreases in knowledge retention were observed in infant growth assessment skills (19.1%) and maternal and antenatal care (18.8%).

An assessment of participants knowledge changes from baseline (pre-intervention 2017) to the 2019 post intervention assessment showed a statistically significant increase of 22.5% in neonatal and child care, and 13.8% in reproductive and sexual health (*p* < 0.001). Interestingly, participants maternal and antenatal care knowledge showed a non-significant decrease from baseline to 2019 (*p* = 0.350). The overall average score also showed a non-significant increase in knowledge (*p* = 0.151).

### 3.3. Effect of Demographics on Participants’ Knowledge and Skills

Stepwise linear regression analysis was used to identify the statistically significant influence of the demographic variables on participants knowledge and skills mean scores in the three assessments (Table 5).

Being female was on average associated with a higher mean score of 11.6% (*p* = 0.033) when responding to reproductive and sexual health questions (β = 11.6), and there was a 14% (β = 14.1) increase in female participants’ infant growth assessment skills scores (*p* = 0.044) compared to males in the 2017 pre-intervention assessment. It was also associated with a higher mean score of 12% in maternal and antenatal care (β = 12.4) in the 2017 post intervention assessment. Participants with prior MCH exposure were on average, associated with a higher mean score of 25% (β = 25.4) and 15% (β = 15.2) in neonatal and child care in the 2017 pre-intervention and 2019 post intervention assessments respectively. Having an African language as a first language was associated with a lower mean score of 16% (β = −16.0) and 16.6% (β = −16.6) in participants’ infant growth assessment skills mean scores in the 2017 pre-intervention and 2019 post intervention assessments respectively. Fourth year locum experience did not have any significant influence on participants’ knowledge and infant growth assessment skills scores.

## 4. Discussion

To our knowledge, this is the first study of its kind to develop an integrated framework for undergraduate pharmacy training in MCH, which combined different components such as reproductive and sexual health, maternal and antenatal care, neonatal and child care, and infant growth assessment skills. The literature collected at the time of the publication of this work showed that most studies of this nature in Pharm D or Bachelor of Pharmacy programs focused on assessing either knowledge, perception, or curriculum content of single components of MCH, such as contraception [27,28,29,30], preconception care [31], immunization [32,33,34], pregnancy and breastfeeding [35,36], and pharmaceutical care for pediatrics [37] which is consistent with the traditional fragmented approach to teaching MCH.

This study aimed to develop and implement a framework for training undergraduate pharmacy students in MCH to ensure that graduating pharmacists are equipped with the required knowledge and skills.

### 4.1. Knowledge and Skills Assessment

Being a female was significantly associated with an increase in participants’ mean score in reproductive and sexual health skills in our study. This may explain the above average mean scores observed for this component in our 2017 pre-intervention (baseline) assessment as 77% of the participants were females of reproductive age with post-high school educational exposure. This observation is supported by a cross-sectional survey of 244 rural Sierra Leone women, where 55.1% of the participants had never been to a school, and 1.2% had tertiary education [38]. The knowledge and reported practices of the women showed that those with high school or tertiary education had better general knowledge of MCH practices compared to others with primary or no formal education. Participants in the Sierra Leone study had an overall health knowledge mean score of 61.6%, although their knowledge of some health guidelines was insufficient. An awareness of the baseline knowledge of the participants in this study would enable the teaching staff to pitch the lecture content at a level that is appropriate and beneficial for students’ learning. Despite participants’ knowledge being above average at baseline in this study, it does not support the incidences of teenage pregnancy experienced in South Africa, where more than 20% of girls aged 15–19 years report being pregnant at least once [39]. This further supports the need for the inclusion of the appropriate level of MCH content in health professionals’ training.

Generally, participants’ mean scores for the MCH components and overall mean scores for the two assessments was above the university’s pass mark of 50% in the 2017 and 2019 post intervention assessments, with higher mean scores than the baseline assessments. This was similar to the results observed by Zaman and Rauf (2011) following a post intervention assessment after students’ exposure to an integrated MCH module [14].

The observed increase in knowledge acquisition in the 2017 post intervention (eight weeks) assessment indicated that the intervention was appropriate and successful. A similar result was obtained in a longitudinal cohort study on medical students’ retention of anatomical knowledge in an integrated problem-based learning curriculum at a college in the United States of America [16]. It was observed that students entered medical school with a low level of anatomy knowledge which increased during the first-year post intervention (10 ± 9 to 46 ± 12%). In our study, the highest mean score for each knowledge component was also recorded in the assessment carried out eight weeks post intervention. This may be attributed to the freshness of the knowledge acquired immediately post intervention since the retention of knowledge potentially decreases over time. A similar result was observed in a study of three cohorts of fourth year medical students in Brazil where the retention of knowledge and clinical skills in basic pediatric cardiology was assessed immediately post intervention and six months and one year later [40]. The highest mean score was recorded in the immediate post intervention assessment with a reduction in the scores in subsequent assessments.

A significant reduction in students’ knowledge retention in the 2019 post intervention assessment (two years later) was observed except in reproductive and sexual health care where the reduction was not statistically significant. An explanation for the non-statistically significant reduction in scores observed in this MCH component may be attributed to the fact that participants had received recent exposure (contraception lectures 2) before the assessment as a follow on to the lectures and practical demonstration of contraceptive products in the second year of study. Additionally, revision of second year content may have preceded the teaching of the fourth-year content which would have refreshed students’ memory, subsequently aiding knowledge retention. El-Ibiary et al. (2018) suggested in their study that using several teaching methods and active lectures, such as practical demonstration workshops and assignments in reproductive and sexual health enhanced students’ knowledge and confidence [41]. Other authors also indicated that a high level of knowledge retention can be achieved through the use of active learning methods and longitudinal reinforcement as a result of content integration across the four years of the study [16,42].

Conversely, the reduction in knowledge retention in the 2019 post intervention assessment can be attributed to non-use or non-practice, which leads to long retention intervals. Historical data suggests that substantial knowledge decay is observed in general education knowledge with 70% retention after 1 year of nonuse; 40–50% after 2 years; 30% after 4 or more years [17]. However, the retention of basic science knowledge from medical school was found to be better depending on frequency of use or reinforcement.

Despite the considerable knowledge increase observed in the 2017 post intervention assessment in the maternal and antenatal care component, which was correlated to the high number of female participants in the study, participants knowledge in the 2019 post intervention study dropped to the 2017 pre-intervention state indicating the high knowledge reduction that occurred in the two-year interval. One of the reasons that may be proffered for this observation may be explained from the result of the study carried out by Peng et al. (2019) where they observed that the adequacy or appropriateness of teacher knowledge transfer significantly influences students’ absorptive capacity and learning outcomes [43]. As such, lecture content and method of teaching must be pitched at a higher level compared to student’s prior knowledge. In addition, participants learning outcomes may be compromised due to the knowledge of the current practice in South Africa where maternal and antenatal care services are provided by nurses at PHC facilities and in the rapidly expanding big chain community pharmacies [44]. This may have detracted from the experience leading undergraduate pharmacy students to believe that they do not have a role in MCH [3].

No baseline assessment was carried out for the infant growth assessment skills component due to timetable constraints (Figure 1). Although the 2017 post intervention assessment took place six months after the initial exposure, the participants recorded a mean score of 74.2%, which indicated adequate infant growth assessment skills and knowledge acquisition and retention. However, the mean score obtained in the 2019 post intervention assessment was significantly reduced to 55.1%. This may be attributed to non-use or insufficient practice of the acquired skills in the time period between the assessments although participants carried out an MCH component in a third-year externship. One of the reasons for the lack of practice would be the subtle elimination of pharmacists from MCH service delivery in pharmacies through the nurse–pharmacist alliances that exist especially in big chain retail pharmacies where most MCH services are provided by nurses [44]. Therefore, undergraduate pharmacy students who work either as post basic pharmacist’s assistants from the second year of study or locum pharmacist’s assistants from the fourth year of study in health facilities may have none or a few privileges of encountering MCH cases. A meta-analysis of the factors that influence skill loss and retention by Arthur et al. (1998) lends credence to the impact of non-use interval on skill retention [45]. He opined that there is a negative or inverse relationship between skills retention and the length of non-practice interval; as one increased, the other would decrease. He carried out a quantitative analysis of the relationship and showed that after more than one year of non-practice interval of a skill, performance reduces to less than 92% of the capability before the non-use interval. In addition, Amaral and Troncon (2013) observed in their study that participants’ clinical and physical examination skills was retained and tended to increase in areas where the opportunity for practicing the skills was more frequent after the initial acquisition in contrast to less frequently used skills such as the interpretation of chest radiographs and electrocardiogram which reduced significantly [40].

However, since 2016, the SAPC has advocated that undergraduate training in MCH should include hands-on practice of infant growth assessment at PHC facilities. This training requirement is in line with the GPP documented baby and child health, and immunization services that pharmacists can offer. To offer these services, the GPP stipulated that a pharmacy must have the road to health booklet for babies, a baby weighing scale, height chart, and tape measure [8]. Pharmacists’ ability to carry out practical, measurement based baby and infant growth assessments will enable them to provide evidence-based advice to parents and carers [3]. This is especially important in sub-Saharan Africa where health care professional shortages are rife and public health care facilities are underfunded and overwhelmed [24]. Ironically, in contrast to the South African scenario, the trend in developed countries with better health care system and lower patient to practitioner ratios, pharmacists are involved in and specializing in MCH service provision [46,47,48].

### 4.2. Limitations and Recommendations

The study was carried out at one pharmacy school in South Africa with a single cohort of participants, and as such, the results may not be generalizable. The result may not be a true reflection of the class performance as purposive sampling was used in the 2019 post intervention assessment (only students who participated in the two previous assessments were included in the study). The same tool was used throughout the study so students may have become familiar with the questions although students were not provided with the answers to the questions at any point. Since the assessments did not count for course grade, the stakes where low for the participants and this may have affected their performance. Infant growth assessment skills were examined in a written assessment rather than the traditional objective structured clinical examination or practical evaluation due to lack of adequate man power. This may undermine the result of that section.

Further studies on factors that may reinforce students’ knowledge and skills in MCH and improve integration of content are recommended. A study on the knowledge and practices of pharmacists in MCH in South Africa is also suggested.

## 5. Conclusions

A framework for an integrated program for undergraduate pharmacy education in MCH was developed and implemented longitudinally across all four years of study. The framework was underpinned by the SAPC GPP, competency standards, high mortality rates and the need to meet the SDG targets by 2030, MCH health workforce shortages, and task sharing. To ensure that adequate knowledge acquisition occurs, lecture content must be appropriately pitched above students’ baseline knowledge. The introduction of new active learning strategies and longitudinal dispersion of curriculum content across a student’s years of study has been shown to aid knowledge retention. In addition, avenues for undergraduate pharmacy students to carry out work-based learning using their MCH knowledge and skills are important to enhance retention.

## Figures and Tables

**Figure 1 pharmacy-09-00163-f001:**
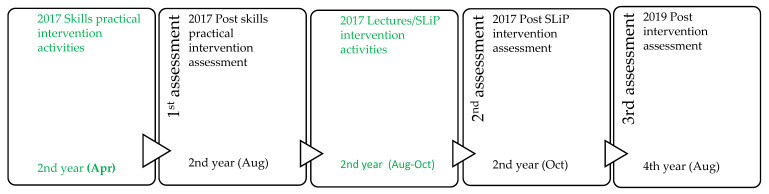
Progression of infant growth skills assessments from the practical intervention (2nd year 2017) to the 2019 post intervention assessment (two years later).

**Figure 2 pharmacy-09-00163-f002:**
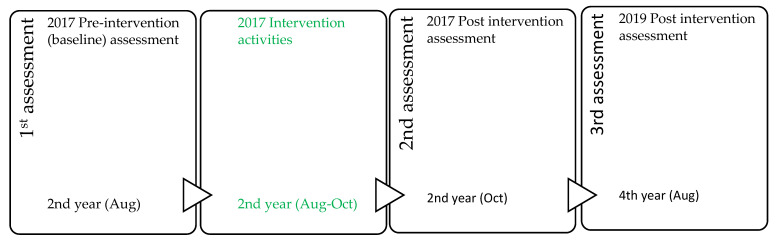
Progression of knowledge assessments from the 2017 pre-intervention to the 2019 post intervention assessment.

**Table 1 pharmacy-09-00163-t001:** The undergraduate pharmacy training integrated MCH framework as aligned with the SAPC competency standards for pharmacists that are relevant to 3 domains: public health, safe and rational use of medicines, and professional and personal practice.

Domain	Competencies	Year Level	Discipline/Module Code	Relevant MCH Knowledge	Skills/SLiP Activity/Duration
Public health	Promotion of health and wellness, medicines information, professional and health advocacy, primary health care	BPharm 1 Semester 2	Introduction to Pharmacology and Clinical Pharmacy (PHC 123)	Environmental and nutritional health (diarrhoeal disease, de-worming, Vitamin A supplementation, pregnancy supplements) (3 h)	SLiP Environmental health visit to underserved community (8 h) ORS salt and sugar dry powder preparation on campus (4 h)
Safe and rational use of medicines and medical devices	Patient consultation, patient counselling, medicines and medical devices safety, pharmacist-initiated therapy, pharmacovigilance	BPharm 2 Semester 1	Pharmacology and Clinical Pharmacy (PHC 213)		Infant growth assessment skills practical on campus (8 h)
		Semester 2	Pharmacology and Clinical Pharmacy (PHC 223)	Pregnancy care, infant carecommunicable diseases, immunization contraception part 1 (15 h)	Contraceptive products practical demonstration (2 h) SLiP-MCH program at a primary health care clinic (9 h)
Professional and personal practice	Patient-centered care, decision-making, collaborative practice, communication	BPharm 3 Semester 2	Pharmacy Practice (PPR 324)		Externship program in a retail pharmacy/health facility (48 h)
Safe and rational use of medicines and medical devices	Patient consultation, patient counselling, medicines and medical devices safety, pharmacist-initiated therapy, pharmacovigilance	BPharm 4 Semester 1	Pharmacy Practice (PPR 414)	Contraception part 2 (2 h)	

Keys: SLiP = service learning in pharmacy, MCH = maternal and child health, ORS = oral rehydration salt.

**Table 2 pharmacy-09-00163-t002:** Demographic data of study participants N = 47.

**Gender**		
Female		36 (77)
Male		11 (23)
**First Language**		
^1^ African languages		23 (49)
^2^ Non-African languages		24 (51)
	**Second Year**	**Fourth year**
**Age (in Years)**		
18 to 30	45 (96)	44 (94)
31 to 45	2 (4)	3 (6)
**Parenting Status**		
No children	47 (100)	46 (98)
**Prior Exposure to MCH**		
Yes	10 (21)	NA
No	37 (79)	NA
**^3^ Locum Experience (in Years)**		
1 to 2	NA	11 (23)
>2	NA	1 (2)

^1^ African languages: All languages of African origin + Afrikaans. ^2^ Non-African languages: English. ^3^ Locum experience: a person who substitutes for another person from the same profession to temporarily fulfil their duties.

**Table 3 pharmacy-09-00163-t003:** Participants’ performance in the individual components and overall assessments relative to the university stipulated pass mark of 50%. The 2017 pre-intervention assessment was a baseline assessment; the 2017 post intervention and 2019 post intervention assessments were carried out eight weeks and two years later, respectively. N = 47.

MCH Components	2017 Pre-Intervention Assessment Mean%, (MD% ^1^, *p*-Value ^2^)	2017 Post Intervention Assessment Mean%, (MD% ^1^, *p*-Value ^2^)	2019 Post Intervention Assessment Mean%, (MD% ^1^, *p*-Value ^2^)
Reproductive and sexual health	56.9 (6.9, 0.005)	73.3 (23.3, 0.000)	70.7 (20.7, 0.000)
Maternal and antenatal care	61.0 (11.0, 0.000)	79.3 (29.3, 0.000)	60.5 (10.5, 0.000)
Neonatal and child care	43.0 (−7.1, 0.016)	76.1 (26.1, 0.000)	65.4 (15.4, 0.000)
Infant growth assessment skills	^4^	74.2 (24.2, 0.000)	55.1 (5.1, 0.187)
Overall average score	53.6 (3.6, 0.045)	75.7 (25.7, 0.000)	62.9 (13.0, 0.000)
Cronbach’s Alpha A (N = 34) ^3^	0.575	0.745	0.776

^1^ Mean Difference—the difference between the university stipulated pass mark of 50% and the mean. ^2^ Significant at *p* < 0.05. ^3^ Number of items in the questionnaire. ^4^ No pre-intervention (baseline) assessment.

**Table 4 pharmacy-09-00163-t004:** Comparison of participants performance between assessments (N = 47).

	Mean Score % for Each Component in the Three Assessments	Mean Difference % between 2017 Pre- and Post Intervention Assessments (*p*-Value)	Mean Difference % between 2017 and 2019 Post Intervention Assessments (*p*-Value)	Mean Difference % between 2017 Pre- and 2019 Post Intervention Assessments (*p*-Value)
Reproductive and sexual health	56.9 ^1^, 73.3 ^2^, 70.7 ^3^	−16.4 (0.000)	2.6 (0.283)	−13.8 (0.000)
Maternal and antenatal care	61.0 ^1^, 79.3 ^2^, 60.5 ^3^	−13.4 (0.030)	18.8 (0.000)	0.5 (0.350)
Neonatal and child care	43.0 ^1^, 76.1 ^2^, 65.4 ^3^	−33.2 (0.000)	10.6 (0.000)	−22.5 (0.000)
Infant growth assessment skills	74.2 ^2^, 55.1 ^3^		19.1 (0.000)	
Overall average score	53.6 ^1^, 75.7 ^2^, 63.0 ^3^	−16.3 (0.000)	12.8 (0.000)	−3.5 (0.151)

^1^ 2017 Pre-intervention assessment. ^2^ 2017 Post intervention assessment. ^3^ 2019 Post intervention assessment.

**Table 5 pharmacy-09-00163-t005:** Stepwise linear regression analysis of the effects of participants demographic variables on knowledge and skills scores.

Assessment	MCH Component			Unstandardized Coefficients		Standardized Coefficients		
		Model		B ^1^	Std. Error	Beta ^2^	T ^3^	*p*-Value ^4^
2017 Pre-intervention	Reproductive and sexual health	1	(Constant)	48.0	4.6		10.4	0.000
			Female	11.6	5.3	0.3	2.2	0.033
	Maternal and antenatal care	1	(Constant)	58.2	11.7		5.0	0.000
			Female	12.3	13.4	0.1	1.0	0.364
			Prior exposure (yes)	−7.9	13.9	−0.1	−0.6	0.573
	Neonatal and child care	1	(Constant)	37.5	2.7		13.8	0.000
			Prior exposure (yes)	25.4	5.9	0.5	4.3	0.000
2017 Post intervention	Reproductive and sexual health	1	(Constant)	75.3	4.9		15.4	0.000
			Female	−4.0	5.6	−0.1	−0.7	0.480
	Maternal and antenatal care	1	(Constant)	70.6	5.3		13.3	0.000
			Female	12.4	6.1	0.3	2.0	0.049
	Neonatal and child care	1	(Constant)	69.0	4.1		17.0	0.000
			Female	7.0	4.7	0.2	1.5	0.140
			Prior exposure (yes)	8.0	4.8	0.2	1.7	0.104
	Infant growth assessment skills	1	(Constant)	65.2	5.9		11.1	0.000
			Female	9.7	6.8	0.2	1.4	0.161
			Prior exposure (yes)	7.3	7.0	0.2	1.0	0.303
		2	(Constant)	73.3	3.6		20.2	0.000
			African language	2.4	6.1	0.1	0.4	0.693
2019 Post intervention	Reproductive and sexual health	1	(Constant)	68.2	4.1		16.5	0.000
			Female	1.7	4.8	0.1	0.4	0.727
		2	(Constant)	73.3	2.4		30.3	0.000
			African language	−7.3	4.0	−0.3	−1.8	0.076
	Maternal and antenatal care	1	(Constant)	55.0	5.7		9.7	0.000
			Female	5.3	6.6	0.1	0.8	0.420
	Neonatal and child care	1	(Constant)	56.9	5.3		10.7	0.000
			Female	6.8	6.1	0.2	1.1	0.269
			Prior exposure	15.2	6.3	0.3	2.4	0.020
	Infant growth assessment skills	1	(Constant)	51.8	7.7		6.7	0.000
			Female	−0.6	8.9	−0.0	−0.1	0.943
		2	(Constant)	61.0	4.5		13.4	0.000
			African language	−16.6	7.6	−0.3	−2.2	0.034

^1^ B column contains the unstandardized beta coefficients that depict the magnitude and direction of the effect on the outcome variable. ^2^ Beta column presents unstandardized beta coefficients for each predictor variable. ^3^ Tolerance column presents values related to assessing multicollinearity among the predictor variables. ^4^ *p*-value is less than 0.05, then that variable has a significant association with the outcome variable.

## Data Availability

The data presented in this study are available on request from the corresponding author. The data are not publicly available due to ethical reasons.

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
