# Peer review of "Development and Implementation of an Integrated Framework for Undergraduate Pharmacy Training in Maternal and Child Health at a South African University"

_pharmacy, 2021, doi:10.3390/pharmacy9040163_

Round 1

Reviewer 1 Report

Development and Implementation of an Integrated 2 Framework for Undergraduate Pharmacy Training in Maternal and Child Health at a South African University

Women’s Health is a global priority currently with organisations such as the WHO and FIP constantly raising awareness that there is a greater need to significantly improve the care of women and girls in both developed and underdeveloped countries alike.

As such developing and implementing curriculum in pharmacy undergraduate programs to develop and train future pharmacists in the specialist area is essential.

I commend the authors in their aspirations in moving towards this goal by developing and implementing relevant curricula.

I do have some reservations with the manuscript and I hope that my comments are seen as useful in developing a manuscript that is easier for the reader to follow and understand.

Introduction

Overall, the introduction sets the scene and relevancy of the topic. There is also a description of why integrated curriculum is aspired to.

Lines 85-87 – consider rephrasing the sentence

Traditional pharmacy department or faculty is generally organized along disciplines such as pharmaceutics, pharmacy practice, 86 pharmaceutical chemistry, pharmacotherapeutics and social pharmacy which inform curriculum organization.

Lines 100-102 – consider rephrasing the sentence

This study is a follow up on a 2017 study that evaluated final year pharmacy students’ knowledge and skills in reproductive, maternal, new- born and child health care at a South African university following  a traditional, fragmented, curriculum content exposure.

 Eg This study follows on from an evaluation conducted in 2017 of final….

At end of the introduction, I suggest outlining clear aims and objectives to help guide the reader in potential outcomes. The manuscript would benefit from a clearer aim and objectives to guide and help the reader. As such review Lines 100-109 to enhance clarity.

Materials and Methods.

This section is difficult to follow and would benefit from major restructuring and review. Can I suggest the use of headings to guide the reader for example:

  • Setting the Scene
  • Participants
  • Data collection
  • Data analysis/Evaluation

Lines131-210 – appear to be a very detailed description of the intervention and the newly designed integrated curriculum.

This is essentially the development of the intervention and could be described as Phase 1. Although this is relevant and interesting for the reader, it is difficult to follow, being very ‘wordy’ .This may benefit from a detailed review to highlight relevant points and provide a summary of the content for each MCH component.

Phase 2 for example would be the evaluation of the MCH components. Here you could describe participants, data collection and analysis.

Lines 113 from the first to fourth year of study

Consider rephrasing  eg - involving students in all four years of the curriculum

Data analysis – Lines 211-227

This would benefit from a review and a more concise description.

I would include the following information in your method.

  • Outline number of MCQs and number of short questions as part of the overall 34 question assessment piece.
  • Also, outline number of questions evaluating knowledge and number evaluating skills. Clearly outline that students required a pass mark of 50% in each section, if this is the case
  • I’m assuming moderation was only undertaken for the short answer questions, if so please make this clear.

Results

The results section is also difficult to follow and would benefit from further work and clarity.

I would outline baseline (pre intervention results) first

Then the main aim of the study is to evaluate any difference in scores post intervention using assessments 2 and 3

As such I would recommend restructure of lines 252- 263

3.1 Participant demographics

The participant who became a parent  in the fourth year of study would be identifiable in this cohort and is an ethical issue. I would recommend that this not be included. Also identified in Table 2

Lines 269 – 292

This would benefit from restructure and a more logical approach to providing some commentary to Table 4. Ensure your key findings are clear. At the moment they are a little lost in the commentary.

Tables 3 and 4

Ensure consistency of % eg Reproductive and sexual health - Table 3, 56.9 but Table 4, 57.0. Please check all.

3.3 Effect of demographics on knowledge and skills

The majority of your participants were female (36) compared to 11 males. Are the authors confident in their conclusions drawn from this analysis? If so although interesting, I would question the relevance. If the authors are keen to include I would ensure that this does not detract from the main aims and objectives of the paper – the impact of developing and implementing curriculum on knowledge and skills to an undergraduate pharmacy cohort.

Discussion

Line 44

This may be attributed to the freshness of the knowledge acquired immediate post intervention as decay had not yet set in.

I would suggest alternative working for ‘decay’

Eg retention of knowledge potentially decreases over time.

Lines 58-60 – Please review phrasing

El-ibiary et al opined in their study that using several teaching methods such as active lectures namely, practical demonstration workshops and assignments in reproductive and sexual health that enhanced students’ knowledge and confidence

 Eg suggested that using several ….

Lines 86-116

This paragraph would benefit from a restructure and condensing the concepts to ensure continued reader engagement.

Limitations

Line 130 - Duplicated wording ‘carried out’

Line 134 – ‘ used in the study’ please remove

Assessment of skills is usually done by practical examination – during experiential placements or OSCEs for example. Skills was evaluated via written examination. Although this was outlined in the limitations – this is not a true reflection of skills evaluation.

Conclusion

Lines 147 - I would suggest - developed and implemented

Lines 149-151 – please review – this sentence does not make sense.

Although students performed well above the university’s pass mark post intervention, longitudinal 150 dispersion of curriculum content across the years of study was found to be an effective way for students to retain knowledge.

In the conclusion I would suggest that there needs to be some reference of how to improve students retention of knowledge and skills.

Reviewer 2 Report

The introduction could be reduced. It is very long to read

How are the authors defining integration? The courses have clinical pharmacists teaching, so how is the integration achieved? Could the authors clarify

4.1 In the discussion section, the authors discuss the decay of the knowledge. If the decay is occurring the how are the authors stating that this intervention can help in practice

what measures are the authors (or instructors) taking to improve retention of this knowledge?

Round 2

Reviewer 1 Report

Thank you for addressing the comments so clearly. well done.

Minor spellings and grammatical errors only to rectify
